# Tumor necrosis factor-α (*TNF-α*) -308G >a promoter polymorphism (rs1800629) promotes Asians in susceptibility to *Plasmodium falciparum* severe malaria: A meta-analysis

**Panida Kongjam[1], Noel Pabalan[1]\*, Phuntila Tharabenjasin[1], Hamdi Jarjanazi[2], Wanna Chaijaroenkul[1], Kesara Na-Bangchang[1,3]\***

**1** Chulabhorn International College of Medicine, Thammasat University (Rangsit Campus), Klongneung, Klongluang, Pathumthani, Thailand, **2** Environmental Monitoring and Reporting Branch, Ontario Ministry of the Environment and Parks, Toronto, Ontario, Canada, **3** Graduate Program in Bioclinical Sciences, Chulabhorn International College of Medicine, Thammasat University (Rangsit Campus), Klongnueng, Klongluang, Pathumthani, Thailand

\* noelpabalan@mail.com (NP); kesaratmu@yahoo.com (KN-B)

## Abstract

The multifactorial pathogenesis of severe malaria is partly attributed to host genes, such as those encoding cytokines involved in complex inflammatory reactions, namely tumor necrosis factor-alpha (TNF-α). However, the relationship between *TNF-α* -308G >A gene polymorphism (rs1800629) and the severity of *Plasmodium falciparum* (*P. falciparum*) malaria remains unclear, which prompts a meta-analysis to obtain more precise estimates. The present meta-analysis aimed to better understand this correlation and provide insight into its association in populations with different ethnicities. Literature search outcomes included eight eligible articles in which *TNF-α* -308G >A polymorphism was determined in uncomplicated malaria (UM) and severe malaria (SM) of *P. falciparum* as represented in the case and control groups. Pooled odds ratios (ORs) and 95% confidence intervals (95% CIs) were estimated in standard homozygous, recessive, dominant, and codominant genetic models. Subgroup analysis was based on ethnicity, *i.e.*, Africans and Asians. The analyses included overall and the modified outcomes; the latter was obtained without the studies that deviated from the Hardy-Weinberg Equilibrium. The significant data also underwent sensitivity treatment but not publication bias tests because the number of studies was less than ten. Interaction tests were applied to differential outcomes between the subgroups. Overall and HWE-compliant analyses showed no significant association between the *TNF-α* -308G >A polymorphism and susceptibility to *P. falciparum* SM (ORs = 1.10–1.52, 95%CIs = 0.68–2.79; $P^a$ = 0.24–0.68). Stratification by ethnicity revealed that two significant associations were found only in the Asians favoring SM for dominant (OR = 1.95, 95% CI = 1.06–3.61, $P^a$ = 0.03) and codominant (OR = 1.83, 95% CI = 1.15–2.92, $P^a$ = 0.01) under the random-effects model, but not among the African populations. The two significant Asian associations were improved with a test of interaction with *P*-value of 0.02–0.03. The significant core

**Data Availability Statement:** The data are available within the manuscript itself and the supplementary information.

**Funding:** KN received financial supports from the National Research Council of Thailand under the Research Team Promotion grant (No. 820/2563), and Thailand Science Research and Fundamental Fund, and Thammasat University (Center of Excellence in Pharmacology and Molecular Biology of Malaria and Cholangiocarcinoma). PK received financial support from Thammasat University Research Fund under Thammasat University Research Scholar. The funders had no role in study design, data collection and analysis, decision to publish, or preparation of the manuscript.

**Competing interests:** The authors have declared that no competing interests exist.

outcomes were robust. Results of the meta-analysis suggest that *TNF-α* -308G >A polymorphism might affect the risk of *P. falciparum* SM, particularly in individuals of Asian descent. This supports ethnicity as one of the dependent factors of the *TNF-α* -308G >A association with the clinical severity of malaria. Further large and well-designed genetic studies are needed to confirm this conclusion.

## Author summary

Host genetic factors play important role in the development and progress of severe malaria due to *Plasmodium falciparum* infection. One of the key factors is the abnormality in the gene that encodes cytokines, particularly tumor necrosis factor-alpha (TNF-α), which are involved in complex inflammatory reactions. However, the relationship between the abnormality of this gene and malaria severity remains unclear. The present meta-analysis aimed to better understand this correlation and provide insight into its association in populations with different ethnicities. The analyses showed no significant association. Stratification by ethnicity revealed that two significant associations were found only in the Asians favoring SM for dominant and codominant, but not among the African populations. Results of the meta-analysis suggest that *TNF-α* -308G >A might affect the risk of *P. falciparum* SM, particularly in individuals of Asian descent. This supports ethnicity as one of the dependent factors of the association between the abnormality of this gene and clinical severity of malaria. Further large and well-designed genetic studies are needed to confirm this conclusion.

## Introduction

Malaria is one of the significant parasitic diseases. Host genetic and environmental components during malaria infection significantly contribute to morbidity and mortality in tropical and subtropical regions around the globe [1,2]. Among the five species of plasmodial parasites in humans, *Plasmodium falciparum* infection is the main causative parasite of severe disease and malaria-related deaths [3,4]. The spectrum of clinical severity of symptoms can range from asymptomatic, uncomplicated, to severe malaria [4,5]. According to the recent World Health Organization Malaria Guidelines, uncomplicated malaria is defined as a patient who has a febrile illness and a positive parasitological test (microscopy or rapid diagnostic test) without features of severe malaria. In contrast, severe malaria involves the central nervous system (cerebral malaria), pulmonary system (respiratory failure), renal system (acute renal failure) and/or hematopoietic system (severe anaemia) [5,6]. There is evidence showing clinical manifestations of malaria and fatality depends on the host's immune status, which is mainly orchestrated by inflammatory cytokine tumour necrosis factor-alpha (TNF-α) [7,8]. In patients with malaria, the cytokine response was found to be responsible for elevated fever symptoms following the onset of blood infection and clinical feature [9]. TNF-α levels has been found to be positively correlated with the severity of malaria [10–12].

TNF-α is produced by macrophages, neutrophils, activated lymphocytes, and natural killer cells [2]. The major role of TNF-α in inflammatory responses is a yardstick of patients' reactivity towards malaria infection [13]. Gene variation can regulate gene transcription, protein expression, and thus the biological function of these cytokines. The *TNF-α* gene lies within a region of ~7 kb in the center of the primary histocompatibility complex locus on the short arm of human chromosome 6 [14]. Most extensive studies investigated the correlation between the

TNF-α gene polymorphism in the -308 promoter region and disease status [15–19]. A point substitution at position –308 (in relation to the *TNFα* transcription start site) of the *TNF-α* gene [20] defines the *TNFα* alleles *1 (*G* at –308) and *2 (*A* at –308). This polymorphism (rs1800629) appears to be important in TNF-α production [21–23]. The circulating TNF-α level was found to be higher in *TNF-α* rs1800629A homozygous (*AA*) compared to *TNFα* rs1800629G homozygous (*GG*) individuals [24]. Moreover, the activation potential of the *TNF-α* rs1800629A allele is more significant than *TNF-α* rs1800629G allele in producing TNF-α [25].

Several case-control groups among primary studies have examined the association between rs1800629 polymorphism and disease severity. However, the results of these studies [26–33] are conflicting, probably because of small sample sizes and low statistical power. Although recent meta-analysis [34] reported a significant association between *TNF-α* -308G >A polymorphism and predisposition to severe *P. falciparum* malaria in the allele and dominant model, there were some obvious limitations in their meta-analytic procedure. First, the subjects in the control group consisted of heterogenous phenotypes, including asymptomatic malaria [35], uncomplicated malaria, and other illnesses [36]. Second, additional five component studies were not included in their meta-analysis even genotype frequencies could be obtained. Furthermore, subgroup analysis was not performed in the different populations. Based on the above reasons, the present meta-analysis on the association of *TNF-α* -308G >A and severe malaria was performed to obtain a more precise estimate. Meta-analysis is helpful for detecting associations that may be obscured in studies of limited sample sizes, particularly in those evaluating rare allele frequency polymorphisms. The current study aimed to update the meta-analysis of the published studies to ascertain whether the polymorphisms of *TNF-α* -308G >A increases susceptibility to severe *P. falciparum* malaria and assess their possible association between ethnically diverse populations.

## Materials and methods

**Case-control definition** Patients with severe *P. falciparum* malaria were included as the study populations (cases). Intervention or exposure was not applied in this case. The control group included patients who manifested uncomplicated malaria's clinical symptoms as described by the World Health Organization [6].

## Publication search and selection of studies

This meta-analysis was performed by following the Preferred Reporting Items for Systematic Reviews and Meta-Analyses (PRISMA) guidelines [37]. We searched the electronic MEDLINE database using PubMed, Scopus, Google Scholar, and MedNar (deep web search engine) for association studies as of April 1, 2023. The search terms applied were "Tumor necrosis factor-alpha", "*TNF-α* -308", "*TNF-α*-308G/A", "polymorphism", "malaria", and "rs1800629", with no restriction of language. The cited references of full-text and review articles were manually scanned to increase additional studies. Inclusion criteria were as follows: (i) original case-control design that examined the associations between *TNF-α*-308G >A polymorphism and risk of uncomplicated or severe *P. falciparum* malaria; and (ii) provision of genotype frequency data that allowed calculation of odds ratios (ORs) and 95% confidence intervals (CIs). Exclusion criteria were (i) review or systematic review articles; (ii) studies that did not cover the polymorphisms *TNF-α* -308G >A or disease in question; (iii) non-human subjects; (iv) studies without controls or studies whose genotype or allele frequencies were unusable or absent; (v) not uncomplicated malaria control type; (vi) not *P. falciparum* malaria; and (vii) articles whose genotype data originated from the same group of the population (duplicated genotype data).

## Data extraction and study quality

Two investigators (PK and NP) independently extracted data. Disagreements were adjudicated by a third investigator (PT) and arrived at a consensus. Key information from the selected articles included: the first author's last name, year of publication, country of origin, ethnicity, age of case and control, genotyping method, and malaria type of case and control. Quantitative features from the articles were also extracted. These included numbers of cases and controls, the sum of which generated the sample size, statistical power, genotype frequencies (*GG*, *GA*, and *AA*), and those of the minor allele and the HWE. Using the application in https://gene-calc.pl/hardy-weinberg-page., the HWE was assessed, and the *P*-value of the controls from Pearson's goodness-of-fit $\chi^2$-square test was reported. The potential impact of HWE-non-compliant studies was examined to ascertain genotyping quality. The Clark-Baudouin scale was used to assess the methodological quality of the included studies [38]. Criteria for the evaluation included *P*-values, statistical power, multiple-comparison correction, comparative sample sizes between cases and controls, genotyping methods, and the HWE. On this scale, the scores of $< 5$, 5–6, and $\geq 7$ were considered low, moderate and high quality, respectively.

## Data distribution and statistical power

The distribution of the data was assessed with the Shapiro-Wilks test [39] using SPSS 20.0 (IBM Corp., Armonk, NY, USA). Gaussian (normal) distribution ($P > 0.05$) warranted descriptive and inferential expressions of the mean ± standard deviation (SD) as well as the parametric approach. Otherwise, the median with interquartile range (IQR) and non-parametric tests were used. The statistical power was assessed using the statistical software G*Power [40], assuming an OR of 1.5 with a genotypic risk level of $\alpha = 0.05$ (two-sided), where power was considered adequate at $\geq 80\%$. With complete linkage disequilibrium for all the *TNF-α* polymorphisms, only the rs1800629 (-308G>A) polymorphism was focused, which best represented all included articles.

## Meta-analysis

The risk for severe *P. falciparum* malaria (using raw data for frequencies) was estimated for each study, and comparing the effects on the same baseline. Pooled ORs and 95% Cis were calculated with a significance threshold of $P \leq 0.05$ (two-tailed). *TNF-α* associations with OR were estimated for each study. The presence of zero genotype values warranted applying the Laplace correction, which involves adding a pseudo-count of one to all data set values [41]. This method could be used in the current study because MAF of all included studies are approximately less than 2%, indicating a low prevalence of mutation probability. Pooled ORs with 95% CIs were calculated for the following genetic models: (a) homozygous: *AA* and *GG* genotypes compared with *GG*; (b) recessive: *AA* versus *GA + GG*; (c) dominant: *AA + GA versus GG*; and (d) codominant: *A* versus *G*. Heterogeneity of the study was estimated using $\chi^2$-based Q test, where significance was set at $P \leq 0.10$ [42] and quantified with the measure of variability ($I^2$) statistic [43]. Either the presence of heterogeneity or not warranted using random and fixed-effects models of analyses, respectively [44,45].

Subgroup analysis was performed with respect to ethnicity, including Africans and Asians. The probability of differential risk associations between comparisons warranted testing for the presence of interactions [46]. Sensitivity analysis, which involves omitting one study at a time and recalculating the pooled OR, was used to test for the robustness of the summary effects. Tests for publication bias were used to evaluate the influence of small-study specific effects [47] and were applied on comparisons that met two conditions: (i) associatively significant ($P^a < 0.05$), and (ii) $\geq 10$ studies [48].

Data were analyzed using Review Manager 5.3 (Cochrane Collaboration, Oxford, UK), SIG-MASTAT 2.03, and SIGMAPLOT 11.0 (Systat Software, San Jose, CA).

# Results

## Characteristics of the included studies

The flow chart of the study selection was outlined in Fig 1 according to PRISMA guidelines. The initial search of four databases and four search strings yielded 13,087 citations, which were further reduced to 115 after reviewing titles and abstracts and removing duplicates (S1 Table). After full-text evaluation, 107 articles were excluded because of not conforming to the inclusion criteria. A manual search of the cited reference list yielded no additional articles. Table 1 lists the eight case-control studies included in the meta-analysis [26–33]. Of them, five [26,27,29,32,33] were not included (new) in a recent meta-analysis [34]. The year range of the articles was 1994–2019 and comprised 1,448 severe *P. falciparum* malaria cases and 1,371 uncomplicated malaria control. Patients in three studies were Africans (843 cases/718 controls), and five studies were Asians (605 cases/653 controls). The age of the subjects was heterogeneous (children to adults). The mean and SD values of the normally distributed Clark-Baudouin scores (Shapiro-Wilks: $P = 0.139$) were $6.38 \pm 1.69$, with most (62.5%) of the articles scoring $\geq 7$. These values indicate the high methodological quality of the included studies.

Table 2 shows the quantitative traits of the included studies. Sample sizes ranged from 67 to 819 and the statistical powers ranged from 12.4 to 80.1%, in which only one study [26] had power of more than 80%. The mean and SD of the minor allele frequency in Africans ($0.13 \pm 0.02$) were not significantly different (Unpaired t-test, $P = 0.73$) from those in Asians ($0.11 \pm 0.04$). The control group in one article (12.5%) [33] showed a significant deviation from HWE. The Preferred Reporting Items for Systematic Reviews and Meta-Analyses guidelines checklist provides detailed description of this meta-analysis (S3 Table).

## Meta-analysis outcomes

Table 3 shows 16 comparisons between *TNF-α* -308G >A and risk for severe malaria. Two of the 16 were significant ($P^a = 0.01$–$0.03$), both derived from stratification analysis by ethnicity. The overall analysis showed no significant association in the four genetic models (ORs 1.14–1.52, 95% CIs 0.74–2.79, $P^a = 0.24$–$0.55$). A similar result was also observed in the HWE-complaint analysis (ORs 1.10–1.32, 95% CIs 0.68–2.37, $P^a = 0.36$–$0.68$).

In the Asians population, there were two significant associations between the *TNF-α* -308G >A polymorphism and the risk SM in the dominant (OR 1.95, 95% CI 1.06–3.61, $P^a = 0.03$) and codominant model (OR 1.83, 95% CI 1.15–2.92, $P^a = 0.01$) comparisons. However, when one HWE non-compliant study was excluded [33], only codominant model was retained significant association (OR 1.59, 95% CI 1.19–2.25, $P^a = 0.002$) (S2 Table). The heterogeneity of the two main outcomes was significant ($P^b = 0.01$–$0.05$) with a high percentage of variation ($I^2 = 58$–$68\%$). The random-effects model was therefore, employed in the OR calculations.

Fig 2 illustrates the forest plots of the association between *TNF-α* -308G >A gene polymorphism and the severity of *falciparum* malaria showing the differences in outcome between Africans and Asians in the dominant and codominant models.

## Tests of interaction

Interaction tests were calculated on the significant ethnic subgroup outcomes. Table 4 shows that of 2 comparisons subjected to these tests, the two non-significant African effects

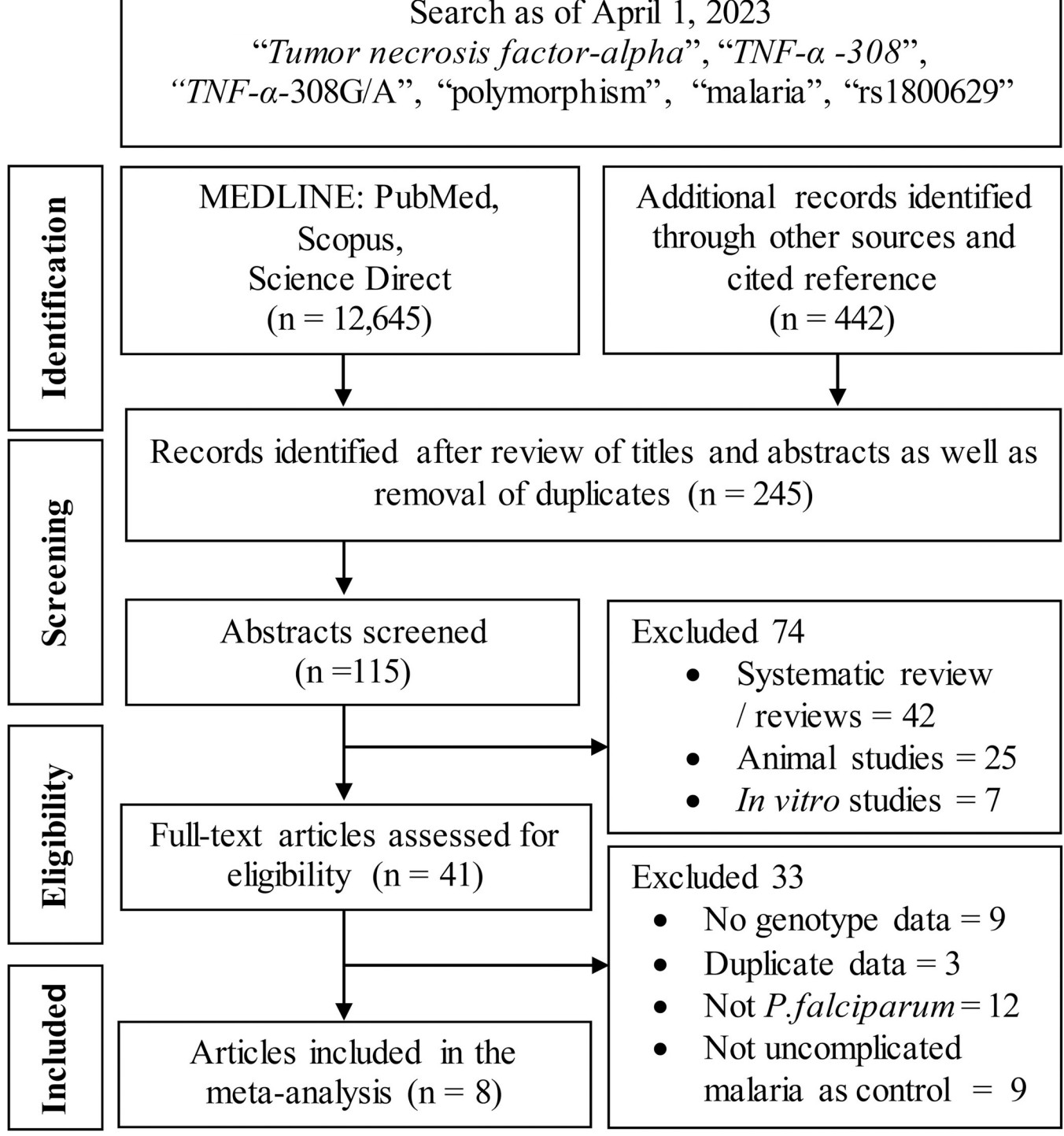

**Fig 1. Flowchart of selection of studies for inclusion in the meta-analysis.**

(dominant: OR 0.72, $P^a$ = 0.30; codominant: OR 0.73, $P^a$ = 0.34) compared with that of the two significant Asian effects (dominant: OR 1.95, $P^a$ = 0.03; codominant: OR 1.83, $P^a$ = 0.01) resulted in significant interaction (dominant: $P_{interaction}$ = 0.023, codominant: $P_{interaction}$ =

**Table 1. Characteristics of the included studies in the *TNF-α* -308G >A polymorphism and its associations with severe *P. falciparum* malaria.**

| | First author | Year | [R] | Country | Ethnic group | Age of patients Case | Age of patients Control | Genotyping method | Cases / Controls | Clark-Baudouin scale |
|---|---|---|---|---|---|---|---|---|---|---|
| 1 | McGuire* | 1994 | [26] | Gambia | African | 3.1 year | 3.9 year | PCR-ASO hybridization | SM / UM | 8 |
| 2 | Meyer* | 2002 | [27] | Gabon | African | Children NM | Children NM | PCR-SSOP | SM / UM | 8 |
| 3 | Olaniyan | 2016 | [28] | Nigeria | African | 46.6 ± 30.7 months | 39.6 ± 25.2 months | Sequenom iPLEX Platform | SM / UM | 7 |
| 4 | Hananantachai* | 2001 | [29] | Thailand | Asian | 25.5 year | 26.25 year | PCR-RFLP | SM / UM | 3 |
| 5 | Mahto | 2019 | [30] | India | Asian | 33.18 ± 13.60 year | 33.60 ± 14.13 year | PCR-RFLP | SM / UM | 7 |
| 6 | Mohanty | 2019 | [31] | India | Asian | NM | NM | Sequencing | SM / UM | 7 |
| 7 | Wattavidanage* | 1999 | [32] | Sri Lanka | Asian | NM | NM | PCR-ASO hybridization and RFLP | SM / UM | 5 |
| 8 | Ubalee* | 2001 | [33] | Myanmar | Asian | NM | NM | PCR-dot-blot hybridization with SSOP | SM / UM | 6 |

*TNF-α*: tumor necrosis factor-alpha; [R]: reference number; *G*: guanine nucleobase; *A*: alanine nucleobase; NM: no mentioned

PCR: polymerase chain reaction; ASO: hybridized with allele-specific oligonucleotides; RFLP: restriction fragment length polymorphism; SSOP: sequence-specific oligonucleotide probes

Age is presented with mean ± standard deviation or an average value

UM: uncomplicated malaria (defined as febrile illness with positive parasitological test and without any of symptom of severe malaria)

SM: severe malaria (see definition in reference no. 6)

* not included in the previous meta-nalysis [34].

0.026) suggesting improved association. This comparative outcome strengthens the statistical evidence favoring the Asian effect.

## Sensitivity analysis and publication bias

Sensitivity analysis was performed using a modified protocol that confined this treatment to the significant findings. Pooled effects that retained ($P < 0.05$) or lost ($P > 0.05$) significance

**Table 2. Details of genotype frequencies of the *TNF-α* -308G >A polymorphism in case and control.**

| | First author | Ethnicity | Sample size Case | Sample size Control | Sample size Total | G *Power (%) | Case (severe malaria) GG-GG | Case (severe malaria) GG-AA | Case (severe malaria) AA-AA | Control (uncomplicated malaria) GG-GG | Control (uncomplicated malaria) GG-AA | Control (uncomplicated malaria) AA-AA | maf | HWE |
|---|---|---|---|---|---|---|---|---|---|---|---|---|---|---|
| | | | 1448 | 1371 | 2819 | | | | | | | | | |
| 1 | McGuire | African | 487 | 332 | 819 | **80.1** | 333 | 135 | 19 | 237 | 89 | 6 | 0.15 | 0.87 |
| 2 | Meyer | African | 98 | 100 | 198 | 28.8 | 81 | 16 | 1 | 74 | 24 | 2 | 0.14 | 0.97 |
| 3 | Olaniyan† | African | 258 | 286 | 544 | 64.3 | 236 | 21 | 1 | 240 | 40 | 6 | 0.10 | 0.29 |
| 4 | Hananantachai† | Asian | 273 | 204 | 477 | 57.8 | 239 | 31 | 3 | 186 | 17 | 1 | 0.05 | 0.15 |
| 5 | Mahto | Asian | 211 | 103 | 314 | 38.2 | 161 | 46 | 4 | 91 | 10 | 2 | 0.07 | 0.40 |
| 6 | Mohanty† | Asian | 40 | 27 | 67 | 12.4 | 27 | 10 | 3 | 14 | 12 | 1 | 0.26 | 0.90 |
| 7 | Wattavidanage† | Asian | 38 | 119 | 157 | 18.7 | 18 | 19 | 1 | 88 | 30 | 1 | 0.13 | 0.37 |
| 8 | Ubalee | Asian | 43 | 200 | 243 | 22.0 | 34 | 7 | 2 | 189 | 7 | 4 | 0.04 | **<0.0001** |

*TNF-α*: tumor necrosis factor-alpha; *G*: guanine nucleobase; *A*: alanine nucleobase; maf: minor allele frequency; HWE: Hardy-Weinberg Equilibrium

(*P*-values where ≤ 0.05 is significant)

†: Laplace correction of genotype frequency

* α = 0.05; OR 1.5. Values in bold indicate statistically powered studies.

**Table 3. Summary effects of the *TNF-α* -308G >A polymorphism with susceptibility to severe *P. falciparum* malaria.**

| | | Test of association | | | Test of heterogeneity | | |
|---|---|---|---|---|---|---|---|
| | *N* | OR | 95% CI | *P*a | *P*b | *I*² (%) | AM |
| **Overall** | | | | | | | |
| Homozygous | 8 | 1.40 | 0.80–2.45 | 0.24 | 0.40 | 3 | Fixed |
| Recessive | 8 | 1.52 | 0.83–2.79 | 0.26 | 0.50 | 0 | Fixed |
| Dominant | 8 | 1.28 | 0.78–2.10 | 0.33 | 0.00001 | 80 | Random |
| Codominant | 8 | 1.14 | 0.74–1.77 | 0.55 | 0.0001 | 78 | Random |
| **HWE-compliant** | | | | | | | |
| Homozygous | 7 | 1.31 | 0.73–2.37 | 0.36 | 0.35 | 11 | Fixed |
| Recessive | 7 | 1.32 | 0.73–2.37 | 0.36 | 0.42 | 0 | Fixed |
| Dominant | 7 | 1.11 | 0.68–1.79 | 0.68 | 0.00001 | 78 | Random |
| Codominant | 7 | 1.10 | 0.72–1.68 | 0.67 | 0.0002 | 78 | Random |
| **African** | | | | | | | |
| Homozygous | 3 | 0.70 | 0.12–3.97 | 0.69 | 0.06 | 65 | Random |
| Recessive | 3 | 0.75 | 0.14–3.89 | 0.73 | 0.07 | 62 | Random |
| Dominant | 3 | 0.72 | 0.39–1.33 | 0.30 | 0.01 | 78 | Random |
| Codominant | 3 | 0.73 | 0.38–1.40 | 0.34 | 0.002 | 84 | Random |
| **Asian** | | | | | | | |
| Homozygous | 5 | 1.93 | 0.75–4.99 | 0.17 | 0.91 | 0 | Fixed |
| Recessive | 5 | 1.82 | 0.71–4.64 | 0.21 | 0.94 | 0 | Fixed |
| Dominant | 5 | **1.95** | **1.06–3.61** | **0.03** | 0.01 | 68 | Random |
| Codominant | 5 | **1.83** | **1.15–2.92** | **0.01** | 0.05 | 58 | Random |

*TNF-α*: tumor necrosis factor-alpha; *G*: guanine nucleobase; *A*: alanine nucleobase

*n*: number of studies; OR: odds ratio; CI: confidence interval; *P*a: *P*-value for association

*P*b: *P*-value for heterogeneity; *I*²: measure of variability; AM: analysis model

Value in bold indicate statistical significance (*P* < 0.05).

were considered robust and not robust, respectively. The results indicate the robustness of two significant outcomes (Table 3). Since there was a limited number of studies (< 10), a test for publication could not be conducted in this meta-analysis.

## Discussion

Examination of the association between -308G >A (rs1800629) and the risk of severe *P. falciparum* malaria relied on the literature's usable data, which paved the way to use the arsenal of meta-analytical processes, including modifier (HWE-compliant), subgrouping, sensitivity, and test of interaction. This is the second meta-analysis to address the association between *TNF-α*—308G >A polymorphisms and the risk of severe *P. falciparum* malaria and the first meta-analysis to unravel this association in different populations. Contrary to the previous meta-analysis [34], the study failed to reveal an association between *TNF-α* -308G >A polymorphism and susceptibility to severe *P. falciparum* malaria in the overall analysis under all genetic models. Moreover, the modifier analyses (excluding the HWE-deviated study) further validated the non-significant association. Interestingly, after stratification by ethnicity, significant associations were found in dominant and codominant carriers in Asians favoring SM, which is the principle finding of the current meta-analysis. Also, both important outcomes showed statistically significant subgroup differences by interaction test as well as the stability of the outcomes.

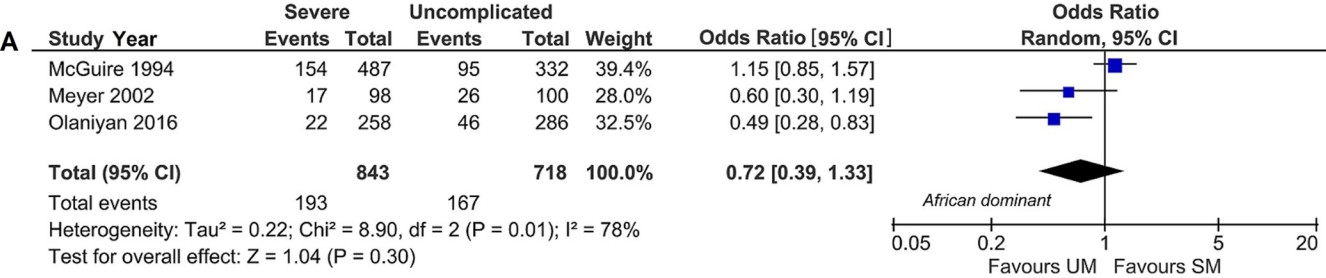

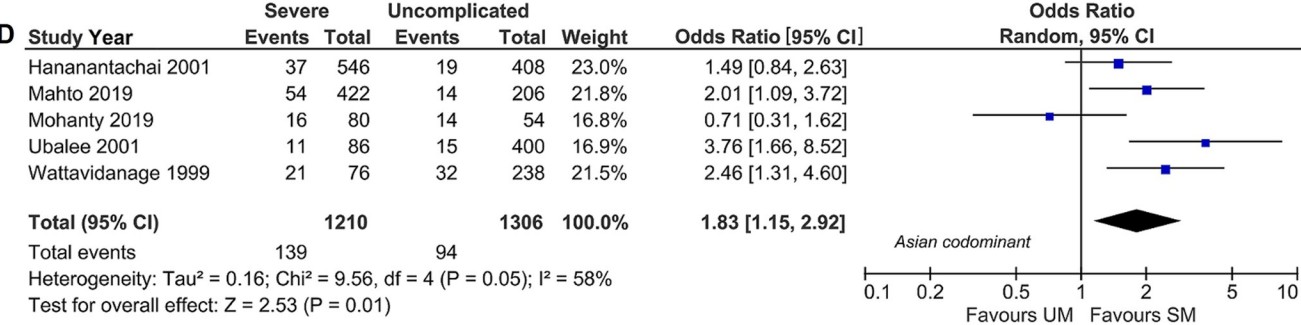

**Fig 2. Forest plot subgroup analysis of association of *TNF-α* -308G >A polymorphism with severe *P. falciparum* malaria in dominant and codominant model.** (A) African dominant (B) Asian dominant (C) African codominant and (D) Asian codominant. CI: confidence interval; *TNF-α*: tumor necrosis factor-alpha; CI: confidence interval; df: degree of freedom; $I^2$: measure of variability; UM: uncomplicated malaria; SM: severe malaria.

## *TNF-α* -308G >A and malaria

The diversity of clinical manifestations of malaria reflects a diverse pathophysiological mechanism depending on factors such as the degree of infection and host immunity [49]. TNF-α is a crucial proinflammatory cytokine in the pathogenesis and control of *P. falciparum* infection

**Table 4. Tests of interactions and sensitivity analysis.**

|  | OR[a] | 95% CI |  | OR[b] | 95% CI |  |  |
|---|---|---|---|---|---|---|---|
| Genetic model | Africans | | | Asians | | P-value of interaction | Sensitivity outcome |
| Dominant | 0.72 | 0.39–1.33 | vs | 1.95 | 1.06–3.61 | **0.02** | Robust |
| Codominant | 0.73 | 0.38–1.40 | vs | 1.83 | 1.15–2.92 | **0.03** | Robust |

OR: odds ratio; a: subgroup with non-significant; b: subgroup with significant; *vs*: *versus*

Value in bold indicate statistical significance at $P < 0.05$.

[30]. Physiologically, the inappropriate or excessive production of TNF-α can contribute to severity and mortality [50,51]. The underlying mechanism can be involved in toxicity and inflammatory processes such as promoting extravasation of neutrophils, lymphocytes, and monocyte to endothelial cells, affecting immune responses by controlling T cell activation. These result in inducing the synthesis of numerous pro-inflammatory cytokines and apoptosis of different cell types, increasing malaria severity as well as other complications [50,51]. The rare allele (*A*) for *TNF-α* (-308G >A) polymorphism produces a higher level of *TNF-α* messenger ribonucleic acid and thus, a high *TNF-α* phenotype, compared to that of major allele *G* [23]. Therefore, the blood TNF-α level based on the genetic background may partly contributable to the intensity of the host immune response.

Previous meta-analysis indicated that patients with UM had higher TNF-α levels than those with asymptomatic malaria [52]. However, there was no difference in TNF-α levels between fatal and non-fatal cases, suggesting that TNF-α levels are not associated with mortality. In turn, the diverse genetic background likely impacts the progression of clinical severity. Moreover, the severity of malaria symptoms may involve the contribution of multi-genes encoding vital host proteins. Previous investigations reported interactions between the TNF-α and other genes [53,54], suggesting that multiple functional polymorphism's effects seem more vital than a single polymorphism's effects. Of note, half of the included articles in the meta-analysis addressed haplotype analysis [27,28,30,32].

The subgroup analysis of the current study showed that Asians significantly favor SM. Meanwhile, Africans favor UM, although there was non-significance in statistics. The finding of different associations is somewhat surprising. A possible explanation for *TNF-α* -308G >A playing a different role in different ethnic could be a result of genetic and clinical heterogeneity between the different populations. Whole-genome linkage studies on severe *P. falciparum* malaria have shown genetic heterogeneity [55]. Although the frequency of *TNFα* -308G >A in Asian populations such as Thailand, Myanmar, Sri Lanka, India, and Vietnam was found to be low (0.06–2.00%), the association between *TNFα* -308G >A polymorphism and malaria clinical outcomes remains evidently clear. The homozygous *AA* of *TNFα* at -308 was associated with severe malaria in Sri Lanka [32], but no association was found in Myanmar patients [33]. As well, it was of interest that different linkage disequilibrium (LD) patterns may contribute to the discrepancy. The polymorphism may be in LD with a nearby causal variant in one ethnic group but not in another. Because Africa is the origin of modern humans, there is a high level of genetic diversity and weak LD in African compared to non-African populations [55]. However, the chance of type I error from multiple testing could not be ruled out. It is unlikely that the discrepancy arises from the number of articles. Fewer studies in Africans (n = 3) generated more total patients in case and control than Asians (n = 5). Further investigation could be performed to strengthen the current results.

## Novelties of the present meta-analysis

Meta-analyses are confronted with a host of epidemiological heterogeneities (different ethnic backgrounds of sample populations) and methodological issues, *e.g.*, diagnostic criteria, inclusion/exclusion criteria, lack of uniformity in how outcomes are measured, poorly defined phenotypes as well as population stratification, which may contribute to difficulties in detecting significant allelic association with phenotypes [56].

Our study delineates the role of *TNF-α* -308G >A polymorphism and the severity of *P. falciparum* malaria, showing there was no statistically significant association for all genetic models. Instead, the significant association was only observed in the racial subgroup. A recent meta-analysis [34] found a significant association between the -308G >A polymorphism and predisposition to the *P. falciparum* clinical severity in the allele comparison and dominant model. There are several possible reasons for such different results. First, the current meta-analysis included five more published studies that were not included in the previous one. In this study, 8 eligible studies with 1,448 cases and 1,371 controls to yield summary statistics, however, Sarangi et al. (2023) analyzed data with 982 cases and 773 controls. Second, the malaria phenotype in the control group was not uniform in previous meta-analyses [34]. The control subjects from two studies [35,36] were asymptomatic malaria and other illness, while the remaining were UM. Third, the present analysis included a subgroup analysis of ethnically diverse populations.

## Strengths and limitations

Pointing out the limitations and strengths contextualizes the interpretation of the present meta-analysis results. There are still some limitations that could be addressed in this study. First, the majority (7/8: 87.5%) of published studies were underpowered (< 80%) to prove authentic associations. Second, the number of patients and studies was not sufficiently large, particularly for subgroup analyses. Third, the present analysis included only English articles. It may have articles in other languages or in local journals. Fourth, the effects of gene-gene and gene-environment interactions were not addressed due to inadequate data. Fifth, the influence of *TNF-α* -308G >A did not preclude effects from other polymorphisms in proximity to *TNF-α* given the complete LD between them. Sixth, publication bias was not analyzed (number of studies < 10), which may have distorted the results. Seventh, the fact that Laplace correction was applied to the genotype frequencies may have distorted our results, possibly highly sensitive modification of a SNP that already has a low MAF. However, this method improves algorithms' exploring power and is not likely affected the results when the variance loss at maximum value of 0.5. Eighth, great disparity of minor allele *A* among the Asians studies may affect the outcome of overall and Asian subgroup. On the other hand, our main results showed that two significant associations were retained in the domianant and codominant model after excluding the highest MAF study (S2 Table). Despite these limitations, the strength of present meta-analyses include (i) the most control group in the selected literature (7/8, 87.5%) were distributed in compliance with HWE, indicating the high genotyping quality of the component studies; (ii) the type of malaria parasite in all patients was homogenous, *i.e.*, *P. falciparum*; (iii) all of the included articles were population-based which facilitates extrapolation of our findings to the general populations in Africans and Asians; (iv) most of the quality of the case-control studies included in the current meta-analysis (7/8, 87.5%) had moderate to high Clark-Baudouin scores indicating good methodological quality; and (v) modifier analysis was performed, which excludes non-HWE compliant study. The outcome of the non-association of *TNF-α* -308G >A polymorphism and SM remains.

## Conclusions

The present meta-analysis suggests that the *TNF-α* -308G >A polymorphisms significantly increased susceptibility to SM in Asians but not in the African population for the dominant and codominant models. The significant association was achieved by various meta-analysis procedures, including modifier (HWE), subgrouping, and test of interaction. Further large-scale and well-designed studies taking into account different ethnic groups remain to be required to confirm these findings.

## Supporting information

**S1 Table. Database search algorithms for *TNF-α* G-308A polymorphism with susceptibility to severe malaria.**
(DOCX)

**S2 Table. Summary effects of the *TNF-α* -308G >A polymorphism with susceptibility to severe malaria in Asian HWE-compliant and excluding Asian study with the highest minor allele frequency.**
(DOCX)

**S3 Table. PRISMA checklist.**
(DOCX)

## Acknowledgments

We thank Dr. Supaporn Kulthinee, Rhode Island Hospital, Brown University, USA, for her assistance in getting some included full-text articles.

## Author Contributions

**Conceptualization:** Noel Pabalan, Phuntila Tharabenjasin, Kesara Na-Bangchang.

**Data curation:** Panida Kongjam, Phuntila Tharabenjasin, Wanna Chaijaroenkul.

**Formal analysis:** Panida Kongjam, Noel Pabalan, Phuntila Tharabenjasin, Hamdi Jarjanazi, Wanna Chaijaroenkul.

**Funding acquisition:** Kesara Na-Bangchang.

**Investigation:** Panida Kongjam, Noel Pabalan, Phuntila Tharabenjasin, Hamdi Jarjanazi, Wanna Chaijaroenkul.

**Methodology:** Noel Pabalan, Phuntila Tharabenjasin, Kesara Na-Bangchang.

**Project administration:** Noel Pabalan, Kesara Na-Bangchang.

**Resources:** Noel Pabalan, Phuntila Tharabenjasin, Wanna Chaijaroenkul.

**Supervision:** Noel Pabalan, Phuntila Tharabenjasin, Kesara Na-Bangchang.

**Validation:** Hamdi Jarjanazi, Wanna Chaijaroenkul.

**Visualization:** Panida Kongjam, Noel Pabalan, Phuntila Tharabenjasin, Hamdi Jarjanazi, Wanna Chaijaroenkul, Kesara Na-Bangchang.

**Writing – original draft:** Panida Kongjam, Noel Pabalan, Phuntila Tharabenjasin.

**Writing – review & editing:** Noel Pabalan, Kesara Na-Bangchang.

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
