## [Decision Letter · Decision Letter 0]

19 Aug 2023

Dear Professor Na-Bangchang,

Thank you very much for submitting your manuscript "Racial disparities in the promoter region G-308A polymorphism (rs1800629) of the tumor necrosis factor-α (TNF-α) gene associated with susceptibility to severe Plasmodium falciparum malaria: a meta-analysis" for consideration at PLOS Neglected Tropical Diseases. As with all papers reviewed by the journal, your manuscript was reviewed by members of the editorial board and by several independent reviewers. The reviewers appreciated the attention to an important topic. Based on the reviews, we are likely to accept this manuscript for publication, providing that you modify the manuscript according to the review recommendations. 

Sincerely,

Walderez O. Dutra, PhD.

Section Editor

Walderez Dutra

Section Editor

Reviewer's Responses to Questions

**Key Review Criteria Required for Acceptance?**

**Methods**

-Are the objectives of the study clearly articulated with a clear testable hypothesis stated?

-Is the study design appropriate to address the stated objectives?

-Is the population clearly described and appropriate for the hypothesis being tested?

-Is the sample size sufficient to ensure adequate power to address the hypothesis being tested?

-Were correct statistical analysis used to support conclusions?

-Are there concerns about ethical or regulatory requirements being met?

Reviewer #1: The methodology is sound and proper statistical analysis methods were used for this meta-analysis study.

Reviewer #2: -Are the objectives of the study clearly articulated with a clear testable hypothesis stated? YES

-Is the study design appropriate to address the stated objectives? YES

-Is the population clearly described and appropriate for the hypothesis being tested? YES

-Is the sample size sufficient to ensure adequate power to address the hypothesis being tested? NA

-Were correct statistical analysis used to support conclusions? NO

-Are there concerns about ethical or regulatory requirements being met? NO

Reviewer #3: The objectives of the study are specifically and clearly to achieve through the study. For the study design is appropriate becuase authors select specifically research articles by using PRISMA guidelines and create excellently inclusion and exclusion criteria leading to have finally the valuable articles (8 out of 13,087). Statistic analysis for this study is appropriate and support the conclusions.

**Results**

-Does the analysis presented match the analysis plan?

-Are the results clearly and completely presented?

-Are the figures (Tables, Images) of sufficient quality for clarity?

Reviewer #1: The results are appropriately presented.

Reviewer #2: -Does the analysis presented match the analysis plan? YES

-Are the results clearly and completely presented? YES

-Are the figures (Tables, Images) of sufficient quality for clarity? YES

Reviewer #3: For the results, authors show the clearly and completely information that they have. They try to explain step by step about association between TNF alpha polymorphism and severity of malaria. Furthermore, they try to go deeply into the subgroup of subjects such as the ethnicity. Lastly, the figures and tables are sufficient quality for clarity. It is easy to understand the points or informations of authors with in a second.

**Conclusions**

-Are the conclusions supported by the data presented?

-Are the limitations of analysis clearly described?

-Do the authors discuss how these data can be helpful to advance our understanding of the topic under study?

-Is public health relevance addressed?

Reviewer #1: The conclusions are supported by the data presented. The limitations of the study are well described. However, the authors do not describe how their findings ccontribute to the understanding and contribution of this polymorphism in complicated malaria among the Asians.

Reviewer #2: -Are the conclusions supported by the data presented? PARTIALLY

-Are the limitations of analysis clearly described? YES

-Do the authors discuss how these data can be helpful to advance our understanding of the topic under study? YES

-Is public health relevance addressed? NO

Reviewer #3: For discussion part, authors try to clearify about association between TNF alpha gene polymorphism and severity of malaria in each ethnic group. They also discuss clearly about limitation and strengths of this study.The conclusion is support clearly the objectives. They try to show the association of severe malaria and Asian population. This point will be knowledge for physician having awareness of severe malaia in Asian population who have TNF alpha polymorphism. So this study have public health relevance addressed.

**Editorial and Data Presentation Modifications?**

Reviewer #1: No recommendation

Reviewer #2: (No Response)

Reviewer #3: Typo is shown in result's part of manuscript (line 271, page 12): "...... the statistical power ranged from 18.7 to 80.1% , in which .................." should be "...... the statistical power ranged from 12.4 to 80.1% , in which .................." .

**Summary and General Comments**

Reviewer #1: This is a nice work of metanalysis and the authors concluded that the polymorphism TNFa -308 is associated with susceptibility among the Asians but not among the Africans. However, my main concern in this study is the minor allele frequency among the Asians. There is a great disparity ranging from 0.04 to 0.26 for the minor allele A among the Asians. Of note. there are two studies from India with completely different minor allele frequency (0.07 and 0.26). It will be interesting for the authors to discuss this disparity and how this could have affected the outcome of the analysis.

Reviewer #2: Major and minor comments follow: 

MAJOR

Please modify the title, as it is it does not reflect what the authors have studied, since the association with ethnicity is based on the stratification studies, no proper analysis for racial disparities between Africans and Asians were compared. 

It is advised to maintain OR estimates and CIs, leave to one decimal case, more than that is not relevant to understand the effect estimate, and it pollutes the manuscript with numbering.

In Methods, line 183, what does criteria vii refers to as duplicated genotype data? Please clarify.

In Methods, line 197, what does corrections for multiplicity refer to? Is it not referring to multiple comparisons’ correction? Please clarify.

In Methods, line 215, please justify the rationale for applicating the Laplace correction in this scenario? rs1800629 has an overall MAF less than 10%, even reaching to 2% in EAS populations according to ENSEMBL. Adding a +1 to genotype counts which are probably to AA genotype is a highly sensitive modification in terms of a SNP that already has a low MAF. Also, having 0 genotype counts doesn’t make inviably the performance and calculation neither of the most relevant genetic models (dominant, codominant), or even the OR. 

Please justify in detail. 

In Results Line 321, the results from Asians and Africans cannot be compared between them, the results for Africans were not significant, so further description that favour UM lack the basis of statistically significant and should not be interpreted. Please remove. 

In Results if the study by Ubalee, 2001 deviated in HWE for the control group there should have been a separate analysis in which the study was excluded from summary statistics, as it has been clearly stated how HWE deviations imply in potential genotyping errors and include it in Supplementary material if the association stands after the exclusion of this study. 

MINOR

In abstract and please review in the rest of the text, when referring to gene and SNP nomenclature modify to TNFA G>A polymorphism or TNFA gene or polymorphism when required. 

In Introduction, line 151 what does UM stands for?

In Materials and Methods please modify the subtitle (line 164) Case-control definition and remove lines 167 -169 that indicate that the meta-analysis followed PRISMA guidelines to include them as the initial line for Publication Search and Study selection in line 172.

Minor Typing and concordance in English redaction errors should be revised. E.g. line 221 in methods correct for heterogeneity.

Line 456. Remove bunch and change for other term.

Reviewer #3: This manuscript has a good composition, harmonization and clarification of writing. Value of meta-analysisof to this study is the usefulness of the association of TNF alpha gene polymorphism and severity of malaria in Asian population. It is valuable information for physician to have more concerned about severe malaria in Asian than other ethnic group.

PLOS authors have the option to publish the peer review history of their article (what does this mean?). If published, this will include your full peer review and any attached files.

Reviewer #1: No

Reviewer #2: Yes: Lucia E. Alvarado-Arnez

Reviewer #3: No

Figure Files:

Data Requirements:

Reproducibility:

References

---

## [Editor Report · Decision Letter 1]

13 Oct 2023

Dear Professor Na-Bangchang,

Thank you very much for submitting your manuscript "Racial disparities in the promoter region G-308A polymorphism (rs1800629) of the tumor necrosis factor-α (TNF-α) gene associated with susceptibility to severe Plasmodium falciparum malaria: a meta-analysis" for consideration at PLOS Neglected Tropical Diseases. As with all papers reviewed by the journal, your manuscript was reviewed by members of the editorial board and by several independent reviewers. In light of the reviews (below this email), we would like to invite the resubmission of a significantly-revised version that takes into account the reviewers' comments. 

We cannot make any decision about publication until we have seen the revised manuscript and your response to the reviewers' comments. Your revised manuscript is also likely to be sent to reviewers for further evaluation.

Sincerely,

Abhay R Satoskar

Section Editor

Walderez Dutra

Section Editor
---

## [Editor Report · Decision Letter 2]

17 Oct 2023

Dear Professor Na-Bangchang,

We are pleased to inform you that your manuscript 'Tumor necrosis factor-α (TNF-α) -308G >A promoter polymorphism (rs1800629) promotes Asians in susceptibility to Plasmodium  falciparum severe malaria: a meta-analysis' has been provisionally accepted for publication in PLOS Neglected Tropical Diseases.

Best regards,

Abhay R Satoskar

Section Editor

Walderez Dutra

Section Editor

---

## [Editor Report · Acceptance letter]

25 Oct 2023

Dear Professor Na-Bangchang,

We are delighted to inform you that your manuscript, "Tumor necrosis factor-α (TNF-α) -308G >A promoter polymorphism (rs1800629) promotes Asians in susceptibility to Plasmodium  falciparum severe malaria: a meta-analysis," has been formally accepted for publication in PLOS Neglected Tropical Diseases.

Best regards,

Shaden Kamhawi

co-Editor-in-Chief

Paul Brindley

co-Editor-in-Chief
